# Tagging motor memories with transcranial direct current stimulation allows later artificially-controlled retrieval

**Daichi Nozaki[1]\*, Atsushi Yokoi[2,3], Takahiro Kimura[4], Masaya Hirashima[5], Jean-Jacques Orban de Xivry[6,7,8]**

[1]Division of Physical and Health Education, Graduate School of Education, The University of Tokyo, Tokyo, Japan; [2]The Brain and Mind Institute, University of Western Ontario, London, Canada; [3]Graduate School of Frontier Biosciences, Osaka University, Suita, Japan; [4]Research Institute, Kochi University of Technology, Kami City, Japan; [5]Center for Information and Neural Networks, National Institute of Information and Communications Technology, and Osaka University, Suita, Japan; [6]Institute of Information and Communication Technologies, Electronics, and Applied Mathematics, Université catholique de Louvain, Louvain-La-Neuve, Belgium; [7]Institute of Neuroscience, Université catholique de Louvain, Louvain-La-Neuve, Belgium; [8]Department of Kinesiology, Movement Control and Neuroplasticity Research Group, KU Leuven, Leuven, Belgium

**\*For correspondence:** nozaki@p.u-tokyo.ac.jp

**Competing interests:** The authors declare that no competing interests exist.

**Abstract** We demonstrate that human motor memories can be artificially tagged and later retrieved by noninvasive transcranial direct current stimulation (tDCS). Participants learned to adapt reaching movements to two conflicting dynamical environments that were each associated with a different tDCS polarity (anodal or cathodal tDCS) on the sensorimotor cortex. That is, we sought to determine whether divergent background activity levels within the sensorimotor cortex (anodal: higher activity; cathodal: lower activity) give rise to distinct motor memories. After a training session, application of each tDCS polarity automatically resulted in the retrieval of the motor memory corresponding to that polarity. These results reveal that artificial modulation of neural activity in the sensorimotor cortex through tDCS can act as a context for the formation and recollection of motor memories.

## Introduction

Context influences memory encoding and retrieval, including fear conditioning responses in rodents (*Maren et al., 2013*) and declarative memory in humans (*Godden and Baddeley, 1975*). Although previous studies have examined whether context-dependency also exists in *motor* memory, context-dependent motor learning based on contextual cues has proven quite difficult. For example, participants have difficulty adapting identical reaching movements to conflicting dynamical environments (e.g., velocity-dependent rightward and leftward force fields) according to a contextual cue (e.g., room color, target color, etc [*Gandolfo et al., 1996*; *Osu et al., 2004*; *Nozaki et al., 2006*; *Hirashima and Nozaki, 2012*; *Kadota et al., 2014*]).

It is only recently that a wide variety of contexts helpful to create distinct motor memories has been discovered (*Osu et al., 2004*; *Nozaki et al., 2006*; *Hirashima and Nozaki, 2012*; *Kadota et al., 2014*; *Cothros et al., 2009*; *Ikegami et al., 2010*; *Howard et al., 2011*; *Yokoi et al., 2011*; *Howard et al., 2012, 2010*; *Sarwary et al., 2013*; *Yokoi et al., 2014*; *Howard et al., 2015*;

**eLife digest** Memory is strongly affected by the context in which a particular memory is formed and remembered. For example, visiting a familiar place can often trigger memories associated or "tagged" with that place. Such tagging also exists for memories related to movement: for instance, distinct motor memories for a limb movement are formed depending on whether the other limb is stationary or moving. However, little is known about how the tagging of such motor memories takes place.

Nozaki et al. have now used a technique known as transcranial direct current stimulation to generate artificial "tags" for motor memories. In the experiments, volunteers tried to move a robotic arm towards a goal while the robot pushed their hand off-course. Sometimes the robot pushed the participant's hand to the left, and sometimes to the right. This makes the task difficult to learn, even when the cue for the direction is provided, as the motor memories that are made to counteract each push overwrite each other.

Nozaki et al. used transcranial stimulation to alter the background electrical activity in the sensorimotor regions of the participants' brains as they performed the robotic arm task. Artificially generating a different pattern of background brain electrical activity for each push direction caused the motor memories associated with leftward and rightward pushes to be tagged differently. Once this association had been learnt, applying the artificial brain stimulation pattern associated with one of the pushes resulted in the participants unconsciously compensating for a push in that direction, even when it was not there.

Overall, the results presented by Nozaki et al. suggest that the background electrical activity seen in the brain can influence how a motor memory is created and later recalled. A future challenge is to investigate whether this technique could be used to help athletes improve their performance or to treat people with movement disorders. Further experiments are also needed to test whether the same approach can influence the formation and recollection of other kinds of memories, such as those related to fear.

*Sarwary et al., 2015*). However, the underlying mechanisms regarding context-dependent motor learning and memory are largely unknown. Notably, contexts that are shown to be useful are often associated with different neural activity patterns of the sensorimotor cortex including the primary motor cortex (M1) and the premotor cortex (PM). For example, distinct motor memories for identical reaching movements can be created depending on whether the opposite arm is stationary or moving (*Nozaki et al., 2006*; *Kadota et al., 2014*; *Nozaki and Scott, 2009*). Consistent with this finding, it has been reported that opposite arm movements alter M1 and PM activity during reaching movements (*Donchin et al., 2001*, *2002*; *Cisek et al., 2003*; *Ganguly et al., 2009*; *Rokni et al., 2003*). In agreement with the significant role of these brain areas in motor learning (*Kadota et al., 2014*; *Gandolfo et al., 2000*; *Li et al., 2001*; *Muellbacher et al., 2002*; *Arce et al., 2010*; *Orban de Xivry et al., 2011*; *2011*; *2013*), we hypothesized that, when a motor memory is formed under different activity patterns in the sensorimotor cortex associated with different contexts, a motor memory specific to this particular activity pattern is created. As a result, later reinstating this activity pattern in these areas should lead to automatic retrieval of the corresponding memory.

Here, we tested this hypothesis directly using transcranial direct current stimulation (tDCS) (*Nitsche et al., 2008*; *Orban de Xivry and Shadmehr, 2014*; *Di Lazzaro and Rothwell, 2014*). tDCS to the sensorimotor cortex modulates spontaneous M1 activity and excitability according to its polarity. As a result, the size of the motor evoked potential induced by transcranial magnetic stimulation is increased or decreased by, respectively, anodal or cathodal tDCS during (*Nitsche et al., 2005*; *2007*) and even after the application of stimulation (*Nitsche and Paulus, 2000*; *2001*; *Siebner et al., 2004*). This suggests that different brain activity patterns (i.e. specific to the polarity of the stimulation) could be artificially created by tDCS. We predicted that motor learning performed under different background activity patterns of the sensorimotor cortex created by tDCS would yield to the formation of separate motor memories. Participants were trained to perform reaching movements in the presence of two conflicting force fields while receiving tDCS on the

sensorimotor cortex. Critically, training for each force field was always associated with a distinct tDCS polarity. We then examined if, after training, applying a particular tDCS polarity reactivated the motor memory associated with this polarity, despite no explicit contextual change.

## Results

Participants performed reaching movements with their right arm in the presence of two conflicting force fields while receiving anodal or cathodal tDCS to the left M1 associated with each force field (training period: 12 blocks of 22 trials). We stimulated M1 because it is known to play a significant role in motor learning (*Kadota et al., 2014*; *Gandolfo et al., 2000*; *Li et al., 2001*; *Muellbacher et al., 2002*; *Arce et al., 2010*; *Orban de Xivry et al., 2011*, *2011*, *2013*). One training block consisted of 18 reaching movements toward a target (movement distance was 10 cm), in either a rightward or leftward velocity-dependent force field (*Shadmehr and Mussa-Ivaldi, 1994*), and four error-clamp trials (first and last two trials of the block, *Figure 1a,c*) to avoid unnecessary learning that could occur during the switch of tDCS polarity. Critically, this polarity (2 mA, bi-hemispheric montage with electrodes on the left and right M1) changed in congruence with the direction of the force field (*Figure 1a,c*) in order to tag motor memories with stimulation polarity. In other words, in this first experiment, anodal and cathodal stimulation of the left M1 were always associated with rightward and leftward force field perturbations, respectively.

During a subsequent test period, we investigated whether modulating activity patterns in the sensorimotor cortex by anodal or cathodal tDCS could induce artificial recollection of the corresponding motor memory. During that period, tDCS polarity alternated between anodal and cathodal, as it did during the training period, but there was no force field perturbation (4 blocks; *Figure 1b,d*). Instead, a series of error-clamp trials was used to evaluate motor memory content throughout the test period (*Figure 1b,d*). If, during training, motor memories were tagged according to tDCS polarity, we should observe leftward force output (this direction was defined as negative) to compensate for the rightward force field during anodal tDCS. On the other hand, application of cathodal tDCS was expected to induce force output in a positive (i.e., rightward) direction. It should be noted that no cognitive and/or contextual cues were provided throughout the experiments.

### tDCS induces motor memory retrieval during the test period

During the test period, the first eight participants received anodal and cathodal tDCS in alternation, starting with anodal stimulation (*Figure 1g*, T-T$_{ACAC}$). In this group, evolution of the force output during the test period (*Figure 2a*, dashed line and open circles) indicates the presence of a tDCS effect. As predicted, the force output exhibited a clear polarity-dependent change, and the forces during cathodal stimulation (blue open circles in *Figure 2a*) were larger than the forces during anodal stimulation (red open circles in *Figure 2a*). Another group of participants (N = 8, T-T$_{CACA}$, *Figure 1g*) were trained with an identical protocol but first received cathodal stimulation during the test period. This subgroup also exhibited clear polarity-dependent effects on force output during the test period (*Figure 2a*, solid line and filled circles), which was consistent with our predictions. In addition, these changes were in anti-phase for the two groups.

Natural exponential decay in force output observed at the beginning of the test period co-occurs with the polarity-dependent effect of tDCS. We took it into account with two different methods. In the first model-free approach, we tried to eliminate this decay effect by contrasting the force output of the two subgroups (*Figure 2* ΔForce obtained by subtracting the data of the ACAC subgroup from the data of the CACA subgroup). This method relies on the assumption that exponential decay was similar in both subgroups. In the second model-based approach, a summation of two exponential curves was fitted to the force output data of each participant during the test period and we analyzed the residuals around the curve in function of tDCS polarity (*Figure 3*). This method bears the advantage of being applied on each subgroup separately.

*Figure 2b* represents the trial-dependent change in the ΔForce calculated using a bootstrap method. If the stimulation had no effect on motor memories, the evolution of ΔForce over the test period should be flat. However, we observed a clear modulation pattern (*Figure 2b,c*): the ΔForce was smaller for the first and third block compared to the second and fourth block. This effect of block order (*Figure 2—figure supplement 1*) on the ΔForce was statistically significant (*Figure 2c*;

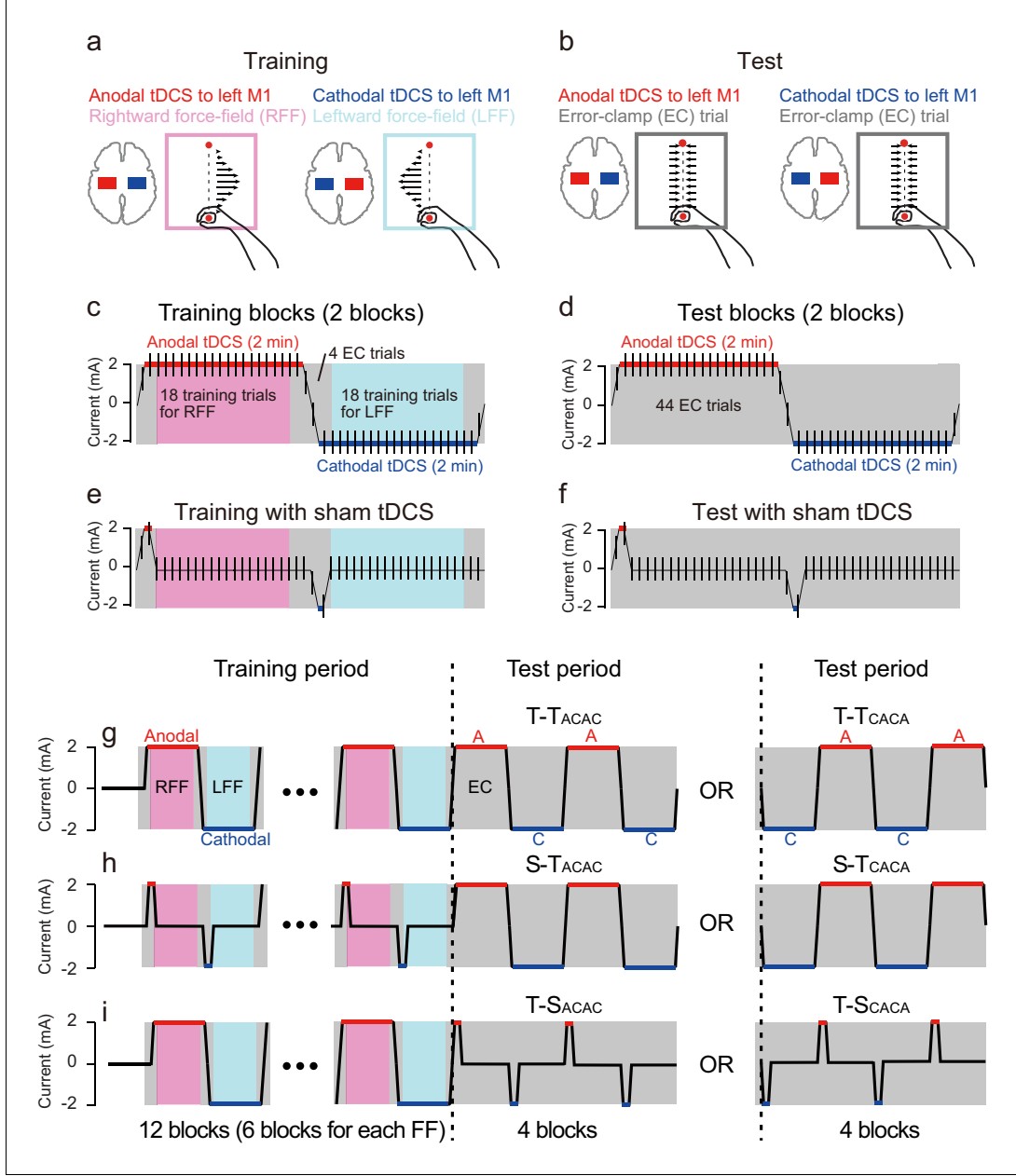

**Figure 1.** Experimental procedure. (a) Participants performed a reaching movement toward a target located 10 cm ahead of the starting position. Participants were trained with a rightward or leftward velocity-dependent force field while receiving anodal or cathodal tDCS to the left M1, respectively. (b) During the test period, the error-clamp trials were used to evaluate how tDCS polarity affected the force output exerted against the force channel. (c) During the training period, anodal (2 mA) or cathodal tDCS (−2 mA) was applied for 2 min, and the transition time between the two was 12 s. When receiving anodal or cathodal tDCS, participants performed 20 reaching movements, and the first and last trials were error-clamp trials. The remaining 18 trials were force field (i.e., training) trials. During the tDCS transition time, 2 error-clamp trials were used to avoid unnecessary motor learning. (d) During the test period, the tDCS pattern was the same as patterns during the training period. However, error-clamp trials were used throughout the entire period. (e,f) In the control groups, sham tDCS was applied during training (e) and the test period (f). In the sham tDCS condition, only the beginning tDCS portion was present. (g) Experimental protocol. After 20 baseline trials, training blocks shown in (c) were repeated 6 times (i.e., total of 12 blocks), and the test blocks shown in (d) were repeated twice (i.e., total of 4 blocks). There was a 3-minute rest period after 8 training blocks were completed. During the test period, the T-T$_{CACA}$ group experienced cathodal tDCS first. (h,i) In the S-T and T-S groups, the training blocks shown in (e), and the test blocks shown in (f) were used, respectively.

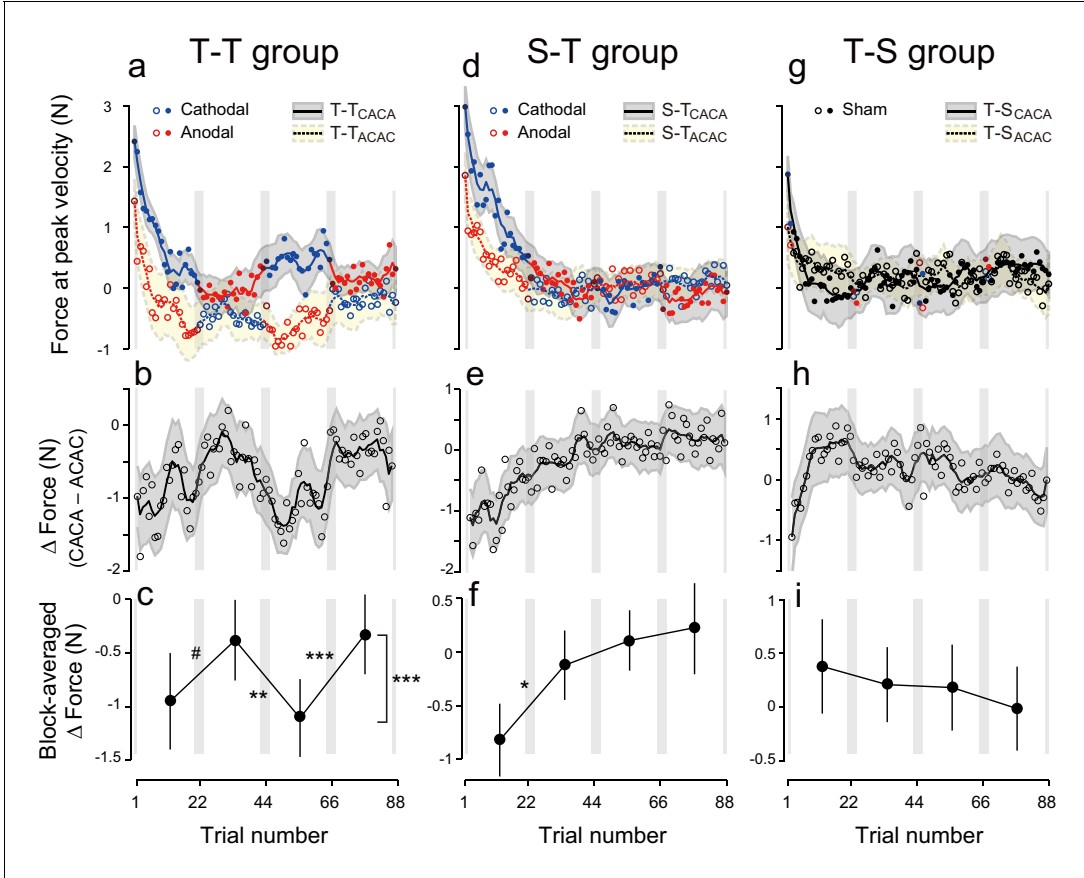

**Figure 2.** Experimental results during the test period. (a) Motor memory evaluated as the force output using the error-clamp trials for the T-T$_{ACAC}$ (open circle) and T-T$_{CACA}$ subgroups (filled circle) during the test period. The positive force indicates rightward force output. Circle color represents tDCS polarity (red and blue for anodal and cathodal tDCS, respectively). The solid and dashed lines are the moving average (5 data points), and the shaded areas indicate standard errors. Gray vertical bars indicate the period during which tDCS polarity changed. Note that the T-T$_{ACAC}$ and T-T$_{CACA}$ subgroups received anodal and cathodal tDCS, respectively, in the first block. (b) ΔForce was calculated as the difference between the T-T$_{ACAC}$ and T-T$_{CACA}$ subgroups in order to reduce the effect of exponential motor memory decay. The bold, solid line and shaded grey area indicates the mean and standard deviation of the bootstrapped samples, respectively (moving average calculated over 5 data points). (c) ΔForce averaged over each block. The mean and standard deviation were obtained from bootstrapped samples. A permutation test was used to test the effect of block order (***p<0.0001 as indicated at the right side). A permutation test was also used to compare the values between the first and second, second and third, and the third and fourth blocks. #p<0.07; **p<0.01; ***p<0.005. (d–i) Results for the S-T group (d–f) and T-S group (g–i).

The following figure supplements are available for figure 2:

**Figure supplement 1.** Definition of factors 'period' and 'block order'.

**Figure supplement 2.** Trial-dependent changes in the handle's peak velocity for the T-T$_{ACAC}$ (open circle) and T-T$_{CACA}$ subgroups (filled circle) during the testing period.

permutation test: p = 0.0009). In addition, this effect appeared stable from the first to the second half of the test period (permutation test: interaction between period and block order: p = 0.313).

Importantly, these changes were consistent with the pattern predicted by the training. For example, in the first block, the force output of T-T$_{ACAC}$ group should be more leftward than that of T-T$_{CACA}$ group and thus the ΔForce was more negative. The degree of ΔForce modulation from the first to second block, from the second to third block and from third to fourth block (the value was defined positive if the change was congruent with the predicted change) was 0.57 ± 0.32 (permutation test: p = 0.0632), 0.73 ± 0.21 (p = 0.0044), and 0.78 ± 0.13 N (p = 0.0001), respectively (mean ± standard deviation of the bootstrapped samples). In our experimental setting, the force output of

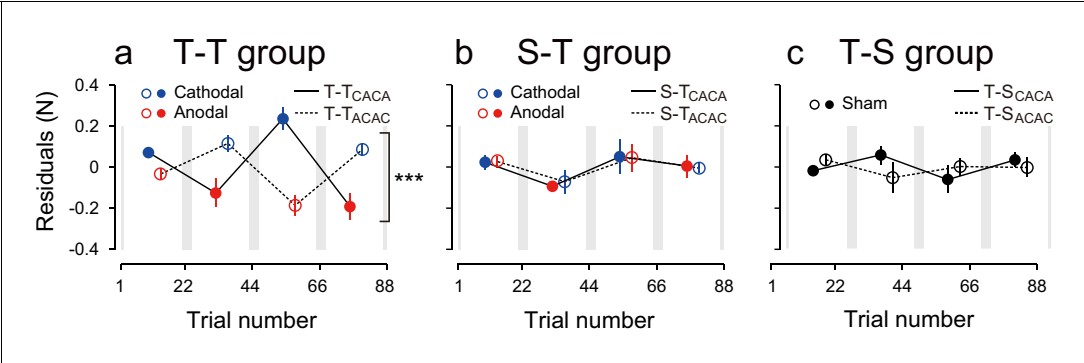

**Figure 3.** Modulation of the residuals, with block, during the test period. (a) The force output residuals from the exponential fitting of 4 blocks for the T-T$_{ACAC}$ (dashed line and open circle) and T-T$_{CACA}$ (solid line and filled circle) groups. The error bars indicate standard errors. Circle color represents tDCS polarity. A 3-way ANOVA indicates a significant interaction between subgroup and block order. \*\*\*p<0.005. (b) Results for the S-T group receiving sham tDCS during the training period. (c) Data for the T-S group receiving sham tDCS during the test period. Circle color was set to black for data obtained when sham tDCS was used.

The following figure supplements are available for figure 3:

**Figure supplement 1.** Calculations for residuals.

**Figure supplement 2.** Polarity-dependent changes in the aftereffect for the T$^{ffrev}$-T groups.

**Figure supplement 3.** The residuals obtained using a single exponential curve instead of two exponential curves.

full adaptation varied from -4 N to 4 N (viscosity was 10 N/(m/s) and the peak velocity of the handle was 0.4 m/s). Thus, approximately 7–10% force modulation was induced via tDCS for the T-T group.

Given the effect of tDCS for muscular force output (*Salimpour and Shadmehr, 2014*; *Tanaka et al., 2009*), one might expect that the movement kinematics were also modulated by the polarity of the stimulation. Such modulation might confound the actual effect of tDCS on force output because the perturbation was a velocity-dependent force-field. However, we did not observe any polarity-dependent changes in movement peak velocity [*Figure 2—figure supplement 2*; interaction between group and block order by repeated measures 3-way ANOVA: F(1,14) = 0.374, p = 0.551, $\eta_p^2$ = 0.024; main effect of block order: F(1,14) = 3.076, p = 0.101, $\eta_p^2$ = 0.18]. This demonstrates that polarity-dependent changes in force output were not caused by hand kinematic modulation due to tDCS.

We also characterized the effect of tDCS by examining variation in force output around the natural exponential decay of motor memories during the test period (model-based approach). If the stimulation had no effect on motor memories, the decay of the force output should simply follow this exponential function and the residuals should be evenly distributed around the exponential curve in all blocks (*Figure 3—figure supplement 1a*). In contrast, if the stimulation polarity influences the force output during the test period, the residuals should be more positive (above the exponential curve) during cathodal stimulation and more negative (below the exponential curve) during anodal stimulation (*Figure 3—figure supplement 1b*). Given that the two subgroups received different polarity during each of the blocks, the polarity-dependent change in residuals was opposite between the two groups (*Figure 3a*). That is, the residuals were always more positive for the group receiving cathodal stimulation during one of the blocks and more negative for the group receiving anodal stimulation during the same block (*Figure 3a*). The modulation of the force output by tDCS was analyzed with a 3-way ANOVA with subgroup as a between-subject factor and period (first or second half of the test period, 2 blocks each) and block-order (first and second block of each half of the test period) as a within-subject factors (See *Figure 2—figure supplement 1* for the definitions of period and block order). Given that the polarity was opposite across groups in function of the block order factor (anodal in the first and third block for T-T$_{ACAC}$ but in the second and fourth block for T-T$_{CACA}$),

the modulation of force output by tDCS resulted in a significant interaction between subgroup and block order [repeated measures 3-way ANOVA: $F(1,14) = 39.24$, $p = 2.08 \times 10^{-5}$, $\eta_p^2 = 0.74$]. For each subgroup separately, there was a polarity-dependent change in the residuals as indicated by the main effect of block order on the residuals [T-T$_{ACAC}$: $F(1,14) = 12.70$, $p = 3.10 \times 10^{-3}$, $\eta_p^2 = 0.47$; T-T$_{CACA}$: $F(1,14) = 28.04$, $p = 1.13 \times 10^{-4}$, $\eta_p^2 = 0.67$]. This effect indicates that, for each subgroup, the force output varied with the polarity of the stimulation.

The modulation of force output with tDCS polarity was also examined when the association between tDCS polarity and force direction was reversed (T$^{ffrev}$-T group), i.e., when anodal and cathodal stimulations were associated with leftward and rightward force fields, respectively (*Figure 3—figure supplement 2*). While the effect appeared clear in one of the subgroups (T$^{ffrev}$-T$_{CACA}$), it was absent in the other one (T$^{ffrev}$-T$_{ACAC}$). This mixed effect yielded a non-significant effect of block order on the ΔForce because the permutation test was based on the two groups, but there was still a significant change in ΔForce from the second to third block (*Figure 3—figure supplement 2b,c*; permutation test: $p = 0.0127$).

The polarity dependent modulation was more apparent in the model-based approach because it allowed us to study each group separately. In this case, we found a significant interaction between subgroup and block-order [repeated measures 3-way ANOVA: $F(1,16) = 5.844$, $p = 0.028$, $\eta_p^2 = 0.27$]. This interaction stems from the polarity-dependent changes in force output observed in the T$^{ffrev}$-T$_{CACA}$ subgroup [main effect of block order on the residuals: $F(1,16) = 7.58$, $p = 0.014$, $\eta_p^2 = 0.32$]. This effect was identical to the one observed for the two T-T subgroups (*Figure 3a*). In contrast, for the other subgroup (T$^{ffrev}$-T$_{ACAC}$), there was no clear effect of stimulation polarity during the test period [simple main effect of block-order on the residuals: $F(1,16) = 0.44$, $p = 0.515$, $\eta_p^2 = 0.027$].

Overall, we recorded data from four groups in the active tDCS conditions (i.e., T-T and T$^{ffrev}$-T groups) and our ANOVA results revealed a significant effect in three out of the four groups. Given our statistical power and effect size, this is exactly what would be expected. We compute the probability that there was an effect of stimulation polarity on force during the test period when the effect is observed in 3 out of the 4 groups. To compute this, we used a rationale provided by *Ioannidis (2005)* but adapted this justification to our positive results, as was done by *Lakens and Evers (2014)*. Given a power of 0.8, the probability of observing three significant and one non-significant finding if there is a true effect is as follows (Type 2 error): $0.8 \times 0.8 \times 0.8 \times 0.2 = 0.1024$. Any of the four groups could yield a non-significant finding, so the a-priori likelihood of finding three out of four significant effects is 0.4096. We also needed to find the probability of observing these results if there was no effect (null hypothesis is true). Given a Type 1 error of 0.05 (significance threshold), this probability is as follows: $p = 0.05 \times 0.05 \times 0.05 \times 0.95 = 0.00011875$. Again, because any group could be non-significant, the probability of finding these results if there is no effect is $p = 0.000475$. With these data, we can compute the positive predictive value (PPV) (*Ioannidis, 2005*). This number represents the post-study probability that the effect is true: PPV $= 0.4096/(0.4096 + 0.000475) = 0.998$. That is, the likelihood that there is a true effect despite the fact that one of the four groups yielded a non-significant effect is 99.8%.

## Active tDCS during the training and test periods is required for tagging and retrieval of motor memories

Polarity-dependent modulation in force output could reflect the effect of anodal and cathodal tDCS in facilitating or suppressing force output. However, this is unlikely, as subgroups in which sham tDCS (tDCS applied only at the beginning part of each block) was applied during the training period followed by active stimulation during the test period (S-T group: S-T$_{ACAC}$ and S-T$_{CACA}$ subgroups, N = 8 for each subgroup, *Figure 1e,h*) showed little influence of tDCS polarity on force output during the test period (*Figure 2d*). While we detected a main effect of block order on the ΔForce permutation test: $p = 0.0263$), this effect was reduced over time (permutation test: interaction between period and block order: $p = 0.025$) (*Figure 2e,f*). Indeed, ΔForce did not exhibit significant block-dependent modulation from the second to third block (permutation test: $p = 0.905$) and from the third to fourth block ($p = 0.278$), although this modulation was significant from the first to second block ($p = 0.0132$) (*Figure 2f*). We also compared the strength of

modulation by subtracting the ΔForce of S-T group from ΔForce of T-T group (*Figure 4a*). Although we did not find the expected interaction between group and block order (p = 0.205), this appeared to change over time (permutation test: interaction between block order, period and group: p = 0.0595). Indeed, the modulation of ΔForce from the second to third block and from the third to fourth block was significantly larger in the T-T group than in the S-T group (*Figure 4b*; permutation test: p = 0.0012 and p = 0.0146, respectively). Together, this data revealed that the modulation of the force output of tDCS was larger in the T-T group than in the S-T group from the second block onwards.

The significant change in ΔForce from the first to second block in the S-T group (*Figure 2e,f*) and the absence of the differences between T-T and S-T groups for these blocks (*Figure 4a,b*; permutation test: p = 0.36) indicate that the application of tDCS immediately after training can influence the exponential decay of motor memory in a manner opposite to the conventional effect of tDCS. Namely, cathodal stimulation makes motor memories last longer than anodal tDCS. Alternatively, the observed effect could reflect the possibility that the motor memory was tagged with the brain activity without tDCS and the application of different tDCS polarities during the test period might differently recruit the motor memories. Either way, these effects were not strong enough to induce block-dependent modulation in the later blocks (*Figure 2e,f*; *Figure 4a,b*).

These results were confirmed by the model-based approach. The residuals around the exponential curves in the S-T group did not exhibit any evidence of polarity-dependent modulation of force output [repeated measures 3-way ANOVA: interaction between subgroup and block order: $F(1,14)$ = 0.005, p = 0.945, $\eta_p^2$ = 0.063] (*Figure 3b*). This discrepancy with the model-free approach is related

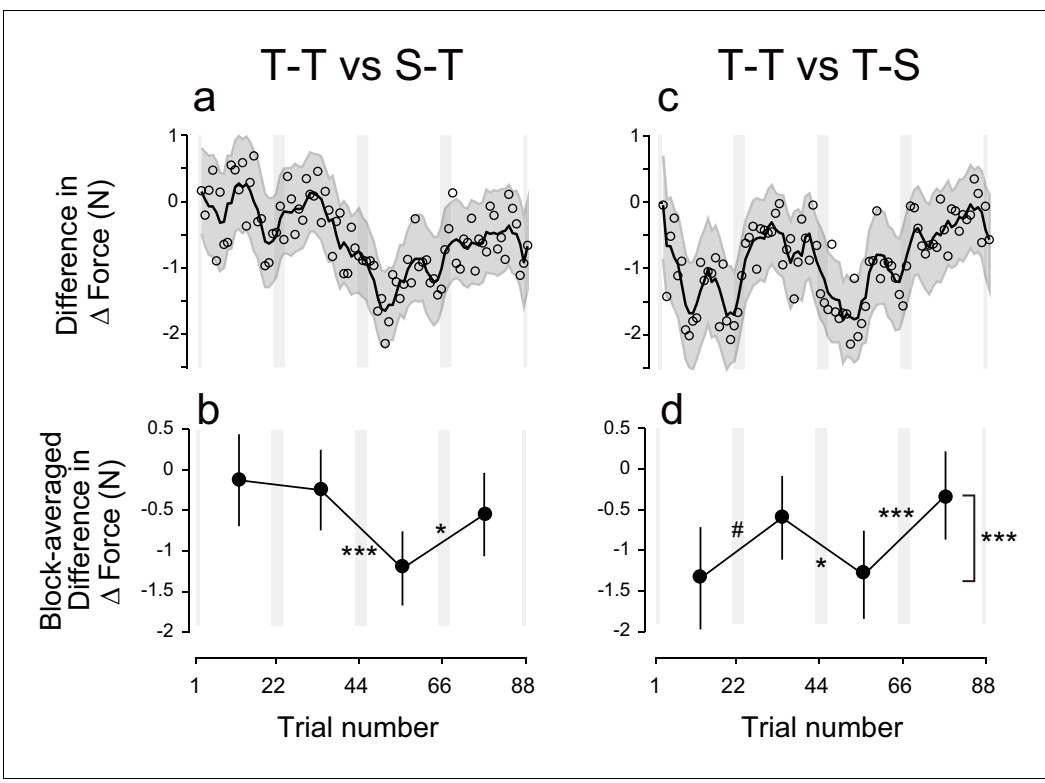

**Figure 4.** Difference in ΔForce between experimental groups. (**a**) ΔForce for the S-T group was subtracted from the T-T group. The solid line and shaded grey area indicates the mean and standard deviation of the bootstrapped samples (moving average calculated over 5 data points). (**b**) The difference in the bootstrap samples of ΔForce was averaged for each block. The error bars indicate the standard deviations of the bootstrapped samples. (**c,d**) Difference in ΔForce between T-T and T-S groups. A permutation test was used to test the effect of block order (***p<0.005 as indicated at the right side of panel (**d**). A permutation test was also used to compare the values between the first and second, second and third , and the third and fourth blocks. #p<0.06; *p<0.05; ***p<0.005

to the ability of the exponential function to capture the early difference in decay rates. The modulation of the residuals via stimulation polarity was also larger for the T-T group than for the S-T group [interaction between group, subgroup, and block order by repeated measures 4-way ANOVA: $F(1,28) = 16.80$, $p = 2.42 \times 10^{-4}$, $\eta_p^2 = 0.37$]. These results confirm that the tDCS effect was restricted to the group where active stimulation was received during the training and test periods.

In another control experiment, active stimulation was applied during the training period while sham tDCS was applied during the test period (T-S group) (*Figure 1f,i*). In this group, we did not observe any polarity-dependent modulation of force output with tDCS polarity (*Figure 2g*). Indeed, the model-free approach applied to ΔForce indicated that there was no significant block-dependent modulation (*Figure 2h,i*; permutation test: main effect of block order: $p = 0.163$). The modulation of ΔForce was significantly larger in the T-T group than in the T-S group (*Figure 4c,d*; permutation test, interaction between group and block order: $p = 0.0018$). In addition, the modulation of ΔForce was significant from the first to second block, from the second to third block, and from the third to fourth block in the T-T group than in the S-T group (*Figure 4d*; permutation test: $p = 0.0573$, 0.0213, and 0.0014, respectively).

These results were confirmed by the model-based approach. For the T-S group, the residuals around the exponential curve were not modulated by tDCS polarity [*Figure 3c*, repeated measures 3-way ANOVA, interaction between subgroup and block order: $F(1,14) = 2.33$, $p = 0.149$, $\eta_p^2 = 0.14$]. In addition, the polarity-dependent modulation of force stimulation observed in the T-T group was significantly larger than the modulation observed in the T-S group [interaction between group, subgroup, and block order by repeated measures 4-way ANOVA: $F(1,28) = 29.79$, $p = 7.93 \times 10^{-6}$, $\eta_p^2 = 0.52$]. Together, these results indicate that activity patterns in the sensorimotor cortex during motor learning needs to be reinstated in order to retrieve those memories.

## PPC tDCS is not effective to tag and retrieve motor memories

Finally, to assess the influence of the electrode position on the polarity-dependent modulation of force output, we performed an additional control experiment in which tDCS was applied to the posterior parietal cortex (PPC: PPC group) (*Figure 5a*). Stimulation of the PPC was unable to yield block-dependent modulation of ΔForce as revealed by the model-free approach (*Figure 5b,c*; permutation test: $p = 0.369$). In addition, block-dependent changes of ΔForce were significantly smaller in the PPC group than in the T-T group (*Figure 5e,f*; permutation test, interaction between group and block order: $p = 0.0082$), although the changes of force output from the first to second (permutation test: $p = 0.0538$) from the second to third ($p = 0.113$) and from the third to fourth block ($p = 0.153$) did not reach the significant level. Similarly, the model-based approach confirmed the absence of effect of tDCS when applied on PPC [*Figure 5d*, repeated measures 3-way ANOVA on the residuals, interaction between subgroup and block order: $F(1,16) = 1.98$, $p = 0.179$, $\eta_p^2 = 0.11$]. In addition, repeated measure 4-way ANOVA indicated that polarity-dependent modulation of force output was significantly greater for the T-T group than for the PPC group [interaction between group, subgroup, and block order: $F(1,30) = 5.10$, $p = 0.031$, $\eta_p^2 = 0.15$], indicating that M1 stimulation was more effective than PPC stimulation in creating and retrieving polarity-dependent motor memories.

## tDCS effects during the training period

We also examined the potential effect of tDCS polarity alternations on the interference between opposing perturbations during the training period (*Figure 6*). To investigate the amount of interference between the two perturbations, we measured the lateral deviations of the first trial during rightward and leftward force-field training. With this measure, higher interference is associated with higher lateral deviation on the first trial. When comparing this measure between the T-T, S-T, and T-S groups, we did not detect any significant differences in lateral deviation during the first trial [repeated measures 3-way ANOVA: $F(2,45) = 2.369$, $p = 0.105$, $\eta_p^2 = 0.088$] (*Figure 6b*). However, when the data from the T-T and T-S groups were grouped together (they received the same training and stimulation during the training period), lateral deviations during the first trial were significantly smaller (i.e., less interference) for groups who received active tDCS whilst training (T-T and T-S groups) than for the sham group (S-T group) [main effect of group: $F(1,46) = 4.435$ $p = 0.041$, $\eta_p^2 =$

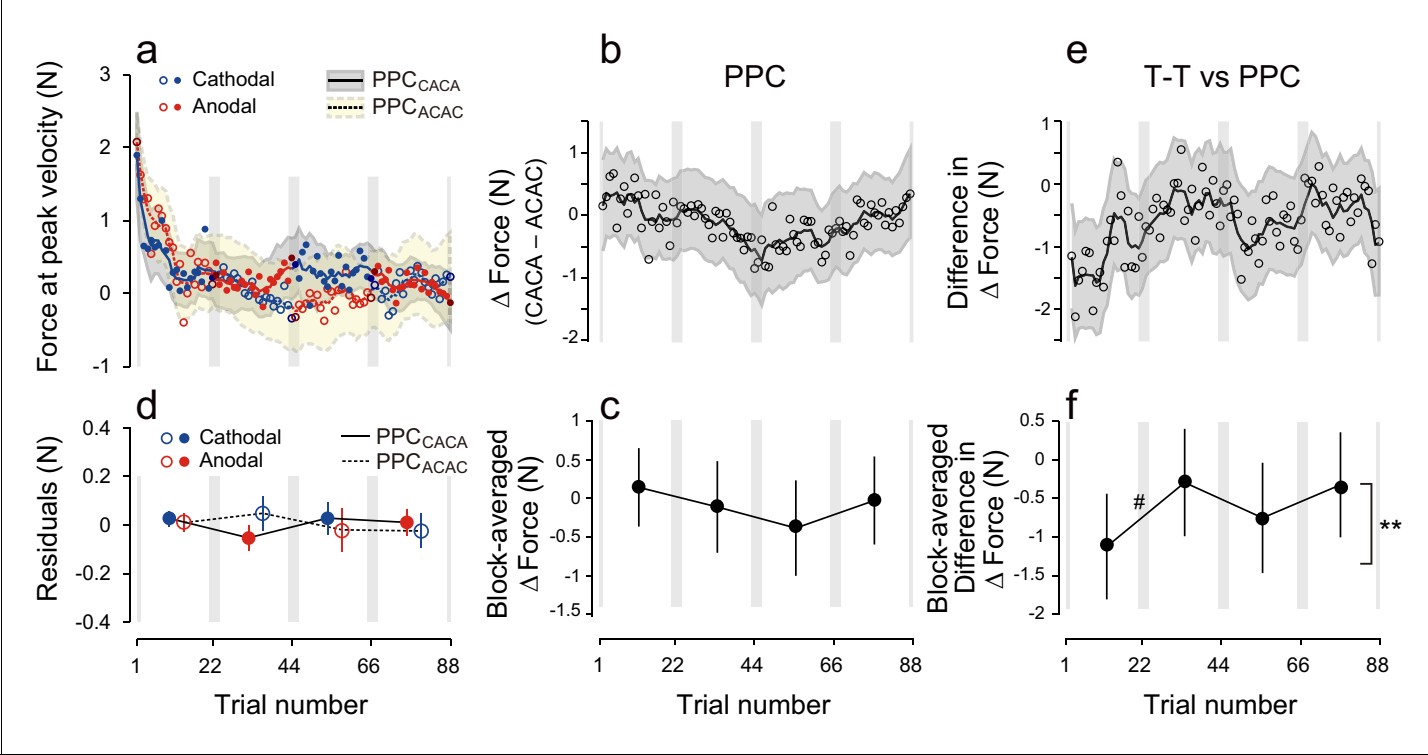

**Figure 5.** Results during the test period for the PPC group. (a) Force output during the test period. (b) Trial dependent change in ΔForce obtained with a bootstrap method. The solid line and grey area indicates mean and standard deviation of the bootstrapped samples (moving average calculated over 5 data points). (c) ΔForce averaged for each block. The error bars indicate standard deviations of the bootstrapped samples. (d) Residuals obtained by subtracting the exponential curves from the original force output. (e, f) Trial-dependent (e) and block-dependent (f) difference in ΔForce between T-T and PPC groups. A permutation test was used to test the effect of block order (**p<0.01 as indicated by the right side of panel (f). A permutation test was also used to compare the values between the first and second, second and third, and the third and fourth blocks. #p<0.06.

0.10; interaction between group and force-field direction (or tDCS polarity): $F_{(1,46)} = 0.148$, $p = 0.708$, $\eta_p^2 = 0.003$]. This suggests that tagging motor memories with different tDCS might have an effect during the training period. However, this result needs to be confirmed in follow-up studies given the limited effect size and borderline significance here. In contrast, lateral deviation at the end of each block was not different between groups [repeated measures 3-way ANOVA, main effect of group: $F_{(1,46)} = 0.574$, $p = 0.453$, $\eta_p^2 = 0.012$; interaction between group and force-field direction: $F_{(1,46)} = 0.401$, $p = 0.53$, $\eta_p^2 = 0.009$] (*Figure 6c*).

## Discussion

Recent studies have demonstrated several striking examples of context-dependent motor learning (*Osu et al., 2004*; *Nozaki et al., 2006*; *Hirashima and Nozaki, 2012*; *Kadota et al., 2014*; *Cothros et al., 2009*; *Ikegami et al., 2010*; *Howard et al., 2011*; *Yokoi et al., 2011*; *Howard et al., 2012*; *Yokoi et al., 2014*; *Howard et al., 2015*). Distinct motor memories for identical movements can be formed and retrieved according to whether the opposite arm is stationary (i. e., unimanual) or moving (i.e., bimanual) (*Nozaki et al., 2006*; *Kadota et al., 2014*; *Nozaki and Scott, 2009*). Motor memories can also be influenced by movement direction of the opposite arm (*Yokoi et al., 2011*; *Howard et al., 2010*; *Yokoi et al., 2014*), whether the movement is discrete or rhythmic (*Ikegami et al., 2010*; *Howard et al., 2011*), or how follow-through movements are performed (*Howard et al., 2015*). Although this type of learning enables us to perform flexible actions for adapting to a wide variety of dynamical environments (*Yokoi et al., 2011*; *2014*), it remains unknown how different behaviors result in the formation of distinct motor memories.

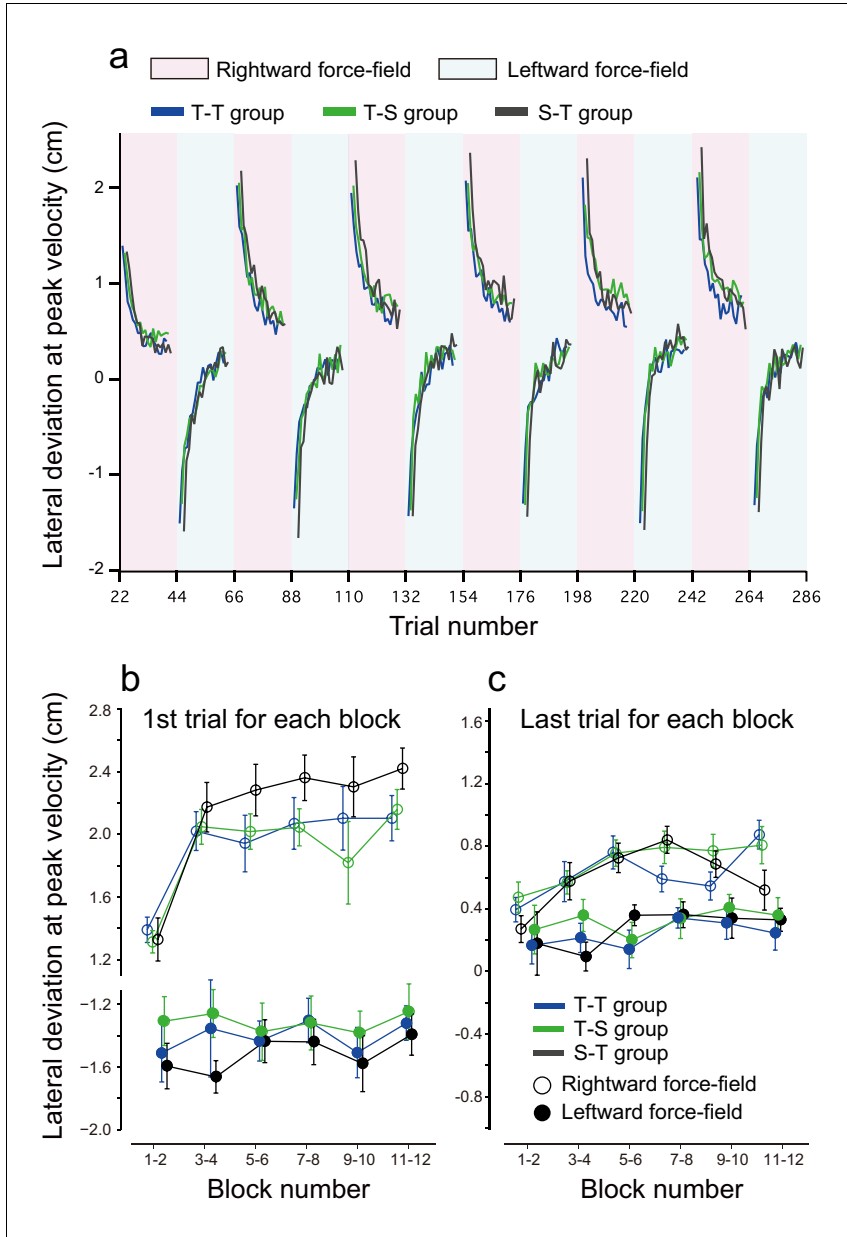

**Figure 6.** Results during the training period. (a) Trial-by-trial changes in lateral deviation evaluated at the peak handle velocity. Only data averaged among participants are displayed; depicting standard errors makes it difficult to observe the data. Red and blue backgrounds indicate the period in which anodal and cathodal stimulation were applied. (b, c) Lateral deviation for the first trial (b) and last trial (c) for each block averaged over participants. The error bar indicates standard error.

Intriguingly, previous neurophysiological and/or brain imaging studies have shown that activity in motor-related neural areas also differs according to how an opposite arm is moving (*Donchin et al., 2001, 2002*; *Cisek et al., 2003*; *Ganguly et al., 2009*), between discrete and rhythmic movement (*Schaal et al., 2004*), or the direction of the follow-through movement (*Baldauf et al., 2008*). It should be noted that previous studies have also suggested that movement adaptation to a novel dynamical environment is accomplished through neural activity changes in these motor-related areas (*Kadota et al., 2014*; *Gandolfo et al., 2000*; *Li et al., 2001*; *Arce et al., 2010*). Thus, it is natural to speculate that these neurons change their activation patterns during motor adaptation, depending on how they are originally recruited within the behavioral context. For example, distinct motor

memories for unimanual and bimanual movements (*Nozaki et al., 2006*) can be explained by considering that partially different populations of M1 and PM neurons are involved for motor adaptation (*Nozaki and Scott, 2009*).

We hypothesized that motor learning performed during distinct background neural activity in the sensorimotor cortex leads to the development of a motor memory that is specifically associated with a particular behavior. We examined this hypothesis by artificially creating two different background activity patterns in the sensorimotor cortex, using anodal and cathodal tDCS, when participants learned to perform reaching movements within 2 conflicting forcefields. Most importantly, each tDCS polarity was always associated with one of two force-fields. Consistent with this hypothesis, our experiments demonstrated that, after training, force output during the test period was modulated by the corresponding polarity (T-T group; *Figure 2a–c*, *Figure 3a*). In contrast, force output modulation was not observed when tDCS was applied to the PPC (PPC group; *Figure 5*). Furthermore, the ability of tDCS to modulate force output, depending on polarity through mere changes in cortical excitability (*Salimpour and Shadmehr, 2014*; *Tanaka et al., 2009*), was insufficient to explain our observed force output modulation (S-T group; *Figure 2d–f*, *Figure 4a,b*).

Participants were also unlikely to modulate the force output according to tDCS-induced skin sensation, as both the PPC and T-T groups experienced similar tDCS sensations throughout the experiments. Taken together, our experiments strongly indicate that differential background activity patterns in the sensorimotor cortex, artificially created when a motor learning task is performed, could lead to the formation of distinct motor memories. These memories could then be artificially retrieved by applying the same-polarity tDCS during a reaching movement.

It is important to emphasize that our experimental paradigm fully departs from conventional tDCS applications, which are traditionally restricted to increasing/decreasing motor output and accuracy, motor memory consolidation, rehabilitation outcomes, and memory functioning (*Nitsche et al., 2008*; *Orban de Xivry and Shadmehr, 2014*). Furthermore, if tDCS can artificially induce motor memory recollection, tDCS may also artificially maintain motor memories when tested in different contexts. Indeed, we have previously shown that reaching movement adaptation to a force-field, while receiving anodal or cathodal tDCS, is more strongly generalized to reaching movements toward another direction performed with different arm postures (*Orban de Xivry et al., 2011*). This improved generalization could be explained by the artificial maintenance of activity patterns in the sensorimotor cortex with tDCS when transitioning from the adaptation phase to the generalization phase.

We assumed that tDCS only influenced neural activity in the sensorimotor cortex, particularly in M1 and PM neurons. Recent studies using fMRI (*Lindenberg et al., 2013*) and electrical fields models (*Gillick et al., 2014*) have indicated that bihemispheric tDCS effects are more localized than when using a conventional stimulation montage (i.e., placing one electrode on the forehead), but we cannot reject the possibility that broader brain areas are also involved. Given our bilateral stimulation, it is also possible that stimulation to the ipsilateral right sensorimotor cortex (ipsilateral to the moving arm) caused the observed effects (rather than the contralateral left sensorimotor cortex). Further studies using a High-Defintion tDCS (*Datta et al., 2009*) will be necessary to clarify these issues. Finally, however, given that tDCS had no effect on memory retrieval when applied on the PPC, the involvement of the underlying parietal areas, including sensory areas, is unlikely or relatively small.

We also assumed that we could create different background activity patterns via anodal and cathodal tDCS. Indeed, previous studies have demonstrated that M1 excitability (*Nitsche et al., 2005*; *Nitsche et al., 2007*) or neuronal firing rates (*Creutzfeldt et al., 1962*) could be modulated according to tDCS polarities. Although the present study only focused on the immediate modulating effect of tDCS on neural activity, tDCS also has long-lasting or plastic effects on the cortex (*Nitsche et al., 2008*), raising questions as to whether activity patterns in the sensorimotor cortex were similar during the training and testing periods. For example, M1 excitability remains elevated, even after anodal tDCS termination, when stimulus duration is sufficiently long (*Nitsche et al., 2007*; *2008*; *Nitsche and Paulus, 2000*; *2001*). However, we considered that any long-lasting effects did not play a dominant role in our results. First, as tDCS polarity switched every 2 min, durations were too short to induce long-lasting effects on cortical excitability. Second, as anodal and cathodal tDCS alternated, any long-lasting effects, including homeostatic plasticity (*Siebner et al., 2004*), were likely cancelled out.

Previous studies have suggested that reinstating the same activity pattern as was present during learning might contribute to one's ability to recall that memory (*Maren et al., 2013*; *Fanselow, 2010*). For example, invasive electrical stimulation to a particular region of the temporal cortex among patients with epilepsy can induce declarative memory recollection. This is perhaps due to reproducing an activity pattern similar to what was observed during voluntary recollection of that same memory (*Jacobs et al., 2012*). Similarly, we used tDCS to reinstate an activity pattern in the sensorimotor cortex during learning in order to induce implicit retrieval of a particular motor memory. These effects are also similar to optogenetics' ability to manipulate a fear memory in mice by directly altering hippocampal neural activity (*Liu et al., 2012*; *Ramirez et al., 2013*).

In summary, our study demonstrates a causal link between background neural activity in the sensorimotor cortex during learning and context-dependent motor memories. Here, we provided initial evidence that human motor memories can be artificially tagged and later retrieved via noninvasive brain stimulation. This manipulation sheds light on the possibility that manipulating the formation and recollection of various memories, including declarative and other types of motor memories (e. g., visuomotor rotation task or sequence learning), can be achieved by artificially changing activity patterns of the corresponding brain region during the learning period. Thus, our novel tDCS application opens up new avenues for implementing tDCS in human memory research.

## Materials and methods

The experiments were conducted in accordance with the Declaration of Helsinki. The ethics committee from The University of Tokyo approved all experimental procedures.

### Participants

We recruited 89 healthy participants (21–42 years old; 52 men and 37 women). Each participant was tested only once. The experiments were terminated for 5 participants (2 men and 3 women) due to impedance failures, strong pain, etc. There were 3 different groups in the main experiments (T-T, S-T, and T-S groups), and each consisted of 2 subgroups [N = 8 (5 men and 3 women) for each subgroup]. Additional control experiments were performed for the $T^{ffrev}$-T and PPC groups. These groups also consisted of 2 subgroups [N = 9 (5 men and 4 women) for each subgroup]. Participants were not given any information regarding to which group they belonged. According to the Edinburg Handedness Inventory, all participants were right-handed (Laterality Quotient: 0.92 ± 0.16), except for 2 women (-1.0 and -0.78). There were no significant differences in the quotient between experimental groups. Prior to the experiments, participants provided informed consent and were paid for their participation.

### Motor task

Participants, sitting on a chair, grasped a robotic manipulandum handle (KINARM End-Point Lab, Bkin Technologies, Canada) with their right hand and moved it horizontally from a starting position toward a target displayed on a mirror placed above the arm. Thus, participants could not directly see their own arm but could see the handle position thorough a white circle (diameter = 1.0 cm) displayed on the mirror. The upper body was fixed to the chair by straps, and a sling was used to hang the forearm horizontally. The start position was located about 20–30 cm from the middle of the chest, and the target was located 10 cm ahead of the start position. The start and target positions were displayed as a circle (diameter = 1.4 cm). After participants maintained their right hand at the starting position for 500–1,000 ms, a green target appeared. After an additional waiting time of approximately 800 ms, the color of the target changed to magenta, indicating that participants should reach towards the target. When the handle reached toward the target, the target turned green. Participants were instructed to make their movements with a peak velocity between 0.35 m/s and 0.45 m/s. The warning 'Slow' or 'Fast' was displayed on the screen if movement speed was outside that range. At the end of the movement, the robot retuned the handle to the starting position. Before the experimental trials, participants performed at least 40 reaching movements (without a force field or tDCS) for practice.

## Force field

Velocity-dependent force fields were imposed on the robotic manipulandum handle during the training period (*Shadmehr and Mussa-Ivaldi, 1994*). The force, $f = (f_x, f_y)$(N), imposed on the handle was always set to be perpendicular to the velocity of the handle, $v = (v_x, v_y)$(m/s), as $f = Bv'$, where $B = (0\ 10;\ -10\ 0)$ and $(0\ -10;\ 10\ 0)$ [N/(m/s)] for the rightward and leftward force field, respectively (x and y directions indicate right-and-left and anteroposterior directions, respectively).

## Error-clamp trials

To evaluate adaptation to the force fields, error-clamp trials (*Scheidt et al., 2000*) were used. During these trials, the handle trajectory was constrained by the robotic manipulandum to a straight line going from the starting position to the target (i.e., a force channel). A virtual spring [15,000 N/m and damper of 100 N/(m/s)] created the force channel. To evaluate motor memory output, the force exerted by participants against the channel was measured. Rightward forces were defined as positive in the present study.

## Experimental procedure

In total, participants performed 373 reaching movements after performing 40 practice trials. Participants performed 20 reaching trials without tDCS or the force field (baseline trials). Twelve trials were error-clamp trials during which the force exerted against the force channel for a baseline was obtained. After the baseline trials, participants performed training trials (training period) followed by test trials (test period).

The training period consisted of 12 blocks of 22 reaching movement trials (*Figure 1*). In each block, the first and last 2 trials were error-clamp trials, and the remaining 18 trials were force field trials. Participants experienced a rightward force field (T-T, S-T, T-S and PPC groups) and a leftward force field (T^ffrev-T group) in the first block. The rightward and leftward force fields alternated in every block. Anodal (2 mA) and cathodal (-2 mA) tDCS to the left M1 (T-T, S-T, T-S and T-T groups) and to the left PPC (PPC group) also alternated in every block. Thus, the rightward force field was learned while receiving anodal tDCS, and leftward force was learned while receiving cathodal tDCS during the training period (T-T, S-T, T-S, and PPC groups). The association between tDCS polarity to M1 and the force-field direction was reversed during an additional control experiment (T^ffrev-T group). Participants were not given any explicit contextual cue for the direction of the force-field throughout the experiments.

In each block, the transition time from 0 mA to 2 mA (or −2mA), and 2 mA (−2 mA) to 0 mA, was 6 s. The reaching target of the first and last trial for each block (error-clamp trial) appeared when tDCS intensity reached 1 mA (or −1 mA). As the target appeared every 6 s (i.e., inter-trial interval was 6 s), tDCS reached 2 mA (or −2 mA) at the second and second-to-last trials for each block (these were also error-clamp trials).

The test period consisted of 4 blocks of 22 reaching movement trials (*Figure 1g–i*). Error-clamp trials were used for all 88 reaching movements. The anodal and cathodal tDCS (± 2 mA) were alternated every block as in the training period. In order to maintain participants' concentration, after 8 blocks of the training period were completed, participants rested for 3 min. After one error-clamp trial without tDCS, the training period was restarted.

The T-T group received active tDCS during both the training and test periods (*Figure 1g*). Training protocols were identical for both subgroups (T-T$_{ACAC}$ and T-T$_{CACA}$ groups). In the odd-numbered blocks (i.e., first, third, fifth, seventh, ninth, and eleventh) and even-numbered blocks (i.e., second, fourth, sixth, eighth, tenth, and twelfth), respectively, participants were trained with the rightward force-field while receiving anodal tDCS and with the leftward force-field while receiving cathodal tDCS (*Figure 1g*). However, during the test period, the block order during which participants received anodal and cathodal tDCS was reversed: T-T$_{ACAC}$ and T-T$_{CACA}$ subgroups received anodal and cathodal tDCS, respectively, in the first block of the test period (*Figure 1g*).

For the S-T$_{ACAC}$ and S-T$_{CACA}$ groups, sham tDCS was used during the training period, but active tDCS was used during the test period (*Figure 1e,h*). Sham tDCS was idential to active tDCS for the initial 9 s of each block(ramp-up for 6 s and constant for 3 s) but ramped down to 0 mA for 6 s (thus, 2 error-clamp trials were performed during these ramp-up and ramp-down phases)

(*Figure 1e*). For the T-S$_{ACAC}$ and T-S$_{CACA}$ groups, sham tDCS was used instead of active tDCS during the test period (*Figure 1f,i*).

In the T$^{ffrev}$-T group, the order of anodal and cathodal tDCS was identical to that of the T-T group, but the leftward force field was imposed on the first block of the training period. Thus, the leftward and rightward force fields, respectively, were always associated with anodal and cathodal tDCS, respectively. The T$^{ffrev}$-T group also consisted of T$^{ffrev}$-T$_{ACAC}$ and T$^{ffrev}$-T$_{CACA}$ subgroups according to the tDCS polarity order received during the test period.

The experimental procedure for the PPC group was the same as for the T-T group, except the PPC was stimulated instead of M1. More specifically, anodal and cathodal tDCS to the left PPC was associated with the rightward and leftward force fields, respectively. The PPC group consisted of PPC$_{ACAC}$ and PPC$_{CACA}$ subgroups according to tDCS polarity order received during the test period.

## tDCS application

An electrical current stimulator applied tDCS (DPS-133A, Dia-Medical Co Ltd, Japan). Two rubber electrodes (5 cm × 7 cm) covered with a sponge soaked in normal saline solution were placed symmetrically at the left and right M1 regions (around C3 and C4; T-T, S-T, T-S, and T$^{ffrev}$-T groups) or at the left and right PPC regions (around P3 and P4; PPC group). We adopted the bihemispheric montage because the stimulated area can be more localized in the sensorimotor cortex, including M1 (*Lindenberg et al., 2013*; *Gillick et al., 2014*). Before applying the electrical current, we carefully checked that resistance between the electrodes was below 5.0 kΩ, via an LCR meter (LCR821, GW Instek, Taiwan), in order to reduce pain or burn injury risk. The values before and after data collection were 3.86 ± 1.09 and 3.38 ± 2.07 kΩ, respectively. During the experiment, the electrical current was continuously monitored using an ammeter (Digital Multimeter CD772, Sanwa, Japan). Participants reported the degree of pain using a numerical rating scale (1–10, 10 indicates maximal pain) after a 3-min break (i.e., after 8 blocks of training were completed) and after the experiment. The reported value was 2.19 ± 1.06 and 2.25 ± 1.08 for the T-T group (mean ± SD for the first 8 blocks and second 8 blocks, respectively), 2.63 ± 1.38 and 3.23 ± 1.43 for the S-T group, and 2.53 ± 1.29 and 2.07 ± 0.81 for the T-S group. It should be noted that participants experienced 4 blocks of sham tDCS in the first 8 blocks for the S-T group and in the second 8 blocks for the T-S group. There was a significant interaction in terms of reported pain between groups and block (i.e., first and second block) [$F(2,44) = 6.876$, $p = 0.003$]. Thus, although the difference in scale values was less than 1, participants might feel stronger skin sensations for active tDCS than for sham tDCS.

## Data analyses

Handle position and exerted force data were sampled at 1000 Hz and then digitally lowpass filtered using a Butterworth filter (cutoff frequency 10 Hz). Handle velocity was obtained by numerical differentiation.

In order to quantify adaptation to the force field during the training period and motor output during the test period, the lateral force exerted by participants against the force channel was evaluated at the handle's peak velocity. Rightward force was defined as positive. We also quantified the handle's lateral deviation at the peak velocity to evaluate learning during the training period. Before analyzing the lateral force and lateral deviation data, baseline trial values were subtracted.

## Statistics

The force output during the test period consisted of natural exponential decay (*Criscimagna-Hemminger and Shadmehr, 2008*; *Brennan and Smith, 2015*) which might obscure the polarity-dependent modulation. We adopted two different approaches to eliminate the effect of the decay.

## Model-free approach

In the first, model-free, approach, we calculated the difference in force between the two subgroups (ΔForce) obtained by subtracting ACAC subgroup data from CACA subgroup data for each group separately:

$$\Delta Force = \overline{F}_{CACA} - \overline{F}_{ACAC},$$

where $\overline{F}$ represents the average force across the corresponding subgroup. This approach was based on the assumption that the decaying pattern was almost identical for both subgroups, because they experienced the same training protocol. To obtain the mean and standard deviation of ΔForce for each group and block separately, we carried out a bootstrap analysis (*Hesterberg et al., 2003*) by randomly sampling the data sets with replacement (N = 10,000) for both the ACAC and CACA subgroups and calculated ΔForce (*Figure 2b, c, e, f, h and i*; *Figure 4*; *Figure 5b, c, e, and f*).

To statistically test the effect of polarity on the ΔForce during the test period for each group separately, we used permutation test (*Hesterberg et al., 2003*) where the variable of interest was the change in ΔForce as a function of the block order where block order refers to the first or second block of each half of the test period (first and third vs. second and fourth) (*Figure 2—figure supplement 1*). We expected ΔForce to be more positive during the second and fourth blocks and more negative during the first and third blocks. The modulation of ΔForce as a function of block order was thus computed as follow:

$$BO = mean(\Delta Force^{b2}) + mean(\Delta Force^{b4})$$

$$-(mean(\Delta Force^{b1}) + mean(\Delta Force^{b3})),$$

where $\Delta Force^{bi}$ represents the $\Delta Force$ values of the i[th] block.

The distribution of this variable of interest under the null hypothesis was obtained by computing all the possible values of the polarity contrast under resampling (N = 10,000) with random reassignment of the subjects in two subgroups (without replacement). The p-value was defined as the portion of the distribution that was more extreme than the observed polarity contrast (*Hesterberg et al., 2003*).

The same technique was used to analyze other variables of interest. The influence of period (first or second half of the test period: *Figure 2—figure supplement 1*) on the block order contrast was quantified by the difference in block order contrast across the two halves of the test period. This interaction was obtained as following:

$$interaction\ between\ period\ and\ block\ order =$$

$$\left(mean(\Delta Force^{b2}) - mean(\Delta Force^{b1})\right)$$

$$-\left(mean(\Delta Force^{b4}) - mean(\Delta Force^{b3})\right),$$

where $\Delta Force^{bi}$ represents the $\Delta Force$ values of the i[th] block.

Block-dependent changes in ΔForce were analyzed similarly. In this case, the change in Δ between consecutive blocks was computed and submitted to the permutation test.

$$\Delta Force^{i \rightarrow i+1} = mean(\Delta Force^{bi+1}) - mean(\Delta Force^{bi}).$$

Statistical test for force between different groups (e.g., T-T vs. S-T groups) was also performed via a permutation test. The observed value corresponded to the difference between the variable of interest of the two groups. For instance, for the block order contrast, the observed value was $BO^{GR1} - BO^{GR2}$ where GR1 and GR2 are the two compared groups. In this case, random reassignment of subjects was performed across groups.

## Model-based approach

In addition to the above-mentioned model-free approach, we used a model-based approach where we estimated the exponential decay for each participant individually and analyzed the residuals around the exponential curve. We reasoned that if stimulation polarity had no effect, the force measures should be evenly distributed around the exponential fit (*Figure 3—figure supplement 1a*). In contrast, if tDCS polarity influences the forces, these residuals should oscillate around that exponential curve across the blocks (*Figure 3—figure supplement 1b*). For each participant, we fitted 2 exponential functions to the force data,

$$y = A_1 \exp(B_1 n) + A_2 \exp(B_2 n) + C,$$

where n represents trial number, and *A*, *B*, and *C* are free parameters. To this end, the 'nlinfit' function in MATLAB was used. We used the 2 exponential functions for data fitting, since a residual calculated using only 1 exponential function could potentially demonstrate artificial polarity-dependent modulation due to data drift. However, our results were not substantially influenced, even when 1 exponential function (i.e., $y = A \exp(Bn) + C$) was used (*Figure 3—figure supplement 3*).

Residuals between the actual data and exponential functions were averaged for each block (the first to fourth blocks) of the test period and tested using ANOVA. The test period consisted of two repetitions of anodal-cathodal tDCS (ACAC subgroup) or cathodal-anodal tDCS (CACA subgroup). To test the effect of subgroup and stimulation polarity on the residuals of the force output during the test period, a repeated measures 3-way ANOVA was conducted with within-subject factors period (the first 2 blocks or last 2 blocks of the test period) and block order (the first or second block for each half of the test period, during which the same tDCS polarity was received in both halves of the test period) and between-subject factor subgroup (ACAC or CACA) (*Figure 2—figure supplement 1*). In this ANOVA, the factor block order can be assimilated with stimulation-induced polarity-dependent changes in force output. Indeed, tDCS polarity was identical between the first and third blocks of the test period (where the block order factor has the same value) and different during the second and fourth blocks of the test period (where the block order factor had another value). Given that the association between block and polarity was opposite between subgroups [the first (or second) block for the ACAC subgroup corresponded to anodal (or cathodal) stimulation and vice versa for the CACA subgroup], we expected, if tDCS influenced the force output, a significant interaction between subgroup and block order. In addition, if the force output varies with tDCS polarity in each subgroup, we expect a main effect of block order for each subgroup separately. The same analysis was used for hand velocity during the test period. We used a 4-way ANOVA where the between-subject factor group was added to contrast the effect of active and sham stimulation (T-T vs S-T or T-T vs T-S) and to contrast the effect of stimulation site (T-T vs PPC).

To test the effect of stimulation type on learning during the training period, the lateral deviation of the first training trial for each block, the averaged value of the last two field trials for each block, and the force exerted during the first error-clamp trial at the end of the block were subjected to an ANOVA with polarity (anodal and cathodal) and block (first to sixth) as within-subject factors and stimulation type (T-T and S-T vs. T-S) as a between-subjects factor.

The statistically significant threshold was set at p<0.05 both for the ANOVA and permutation test. For the results of ANOVA, we reported effect sizes (partial eta squared: $\eta_p^2$) as well as F and p-values.

## Acknowledgements

We thank S Scott, J Diedrichsen, P Gribble, A Pruszynski, J Cashaback and S Furuya for their helpful comments and suggestions; K Tanamachi for recruiting participants; I Hidaka for technical assistance with tDCS; and K Abe, Y Shinya and A Munakata for their professional assistance. This study was supported by the NEXT Program (LS034) to DN, by KAKENHI (A26242062) to DN and MH, and by JSPS and FRS-FNRS under the Japan-Belgium Research Cooperative Program to DN and JJO. JJO was supported by a Brains Back to Brussels fellowship from the Brussels Region (Belgium).

## Additional information

### Funding

| Funder | Grant reference number | Author |
| --- | --- | --- |
| Japan Society for the Promotion of Science | KAKENHI A26242062 | Daichi Nozaki Masaya Hirashima |
| Japan Society for the Promotion of Science | NEXT Program, LS034 | Daichi Nozaki |

| | | |
|---|---|---|
| Japan Society for the Promotion of Science | Japan-Belgium Research Cooperative Program | Daichi Nozaki Jean-Jacques Orban de Xivry |
| Fonds De La Recherche Scientifique - FNRS | Japan-Belgium Research Cooperative Program | Daichi Nozaki Jean-Jacques Orban de Xivry |
| Innoviris | Brains Back to Brussels fellowship | Jean-Jacques Orban de Xivry |

The funders had no role in study design, data collection and interpretation, or the decision to submit the work for publication.

### Author contributions

DN, AY, Conception and design, Acquisition of data, Analysis and interpretation of data, Drafting or revising the article; TK, Acquisition of data, Drafting or revising the article; MH, Conception and design, Drafting or revising the article; J-JOdX, Analysis and interpretation of data, Drafting or revising the article

### Author ORCIDs

Daichi Nozaki, http://orcid.org/0000-0002-1338-8337
Atsushi Yokoi, http://orcid.org/0000-0002-7428-3344
Takahiro Kimura, http://orcid.org/0000-0002-5673-1553
Masaya Hirashima, http://orcid.org/0000-0001-8571-8289
Jean-Jacques Orban de Xivry, http://orcid.org/0000-0002-4603-7939

### Ethics

Human subjects: The experiments were conducted in accordance with the Declaration of Helsinki. The ethics committee from The University of Tokyo approved all experimental procedures. Prior to the experiments, participants provided informed consent.

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
