## [Decision Letter]

Thank you for submitting your article "Tagging motor memories with transcranial direct current stimulation allows later artificially-controlled retrieval" for consideration by *eLife*. Your article has been reviewed by three peer reviewers, including Michael Nitsche and Richard Ivry, who is a member of our Board of Reviewing Editors, and the evaluation has been overseen by Timothy Behrens as the Senior Editor.

The reviewers have discussed the reviews with one another and the Reviewing Editor has drafted this decision to help you prepare a revised submission.

Summary:

The reviewers found considerable merit in this paper. The role of context in sensorimotor learning has been the subject of considerable interest in the past few years. While early results emphasized how sensorimotor adaptation was relatively impervious to context (e.g., color cues), recent work has identified a number of conditions in which contextual cues can be quite powerful. The current study takes the question of context in an entirely new direction, at least within the world of human motor learning, asking if modulation of the (endogenous) neural state can serve as a sufficient contextual cue. To this end, the authors use tDCS to establish the context, associating a force field in one direction with anodal tDCS and a force field in the opposite direction with cathodal tDCS. Of special note here is that the participants are completely unaware of the "cue", something that is not (usually) possible in studies involving behavioral manipulations of context.

Essential revisions:

In general, the reviewers found the motivation of the study of the study to be sound, were intrigued by the results, and found the discussion to be appropriate. While our requests for revision are relatively modest, there is one major, indeed critical, issue that needs to be addressed. While the leftward and rightward force fields induce significant changes in performance, the context effect is relatively small (estimate of about 5% of the force difference). In itself, this is not a concern if the effect is robust-that the authors see any effect here is impressive. However, the key analysis seems overly complex with a lot of parameters. The authors fit a 5-parameter function to every noisy individual-subject test-period data set, the result of which is there are over 400 free parameters (5params/subject x [48 subjects in the main experiments + 36 subjects in the controls], or about 80 in each experiment) to "model" the decay curves in the data. The residuals of these models serve as the primary "data" for the main analysis. With this analysis, one would want to see some graphic depiction of the parameters of the individual double-exponential fit. But, as noted below, we believe there is a better solution.

While there is a general concern with the complexity of this analysis, there is an additional statistical issue here. Between subgroup differences in the fits may seem quite different for the anodal-cathodal switching pattern, but the patterns are not statistically independent. Fitting double exponentials may distort the estimate of the true anodal/cathodal switching effect. This is especially concerning if the switching effects are small compared to the size of the double exponential fits (as is the case here).

The reviewers believe that a much simpler analysis should be possible. The authors could simply analyze the differences in forces that are shown in the middle row, left panel of Figure 2. The middle panel (2e, difference for the sham condition) should probably be subtracted from the left panel (2b, the difference for the stimulation condition), and then the mean of each block can be statistically compared.

[Editors' note: further revisions were requested prior to acceptance, as described below.]

Thank you for resubmitting your work entitled "Tagging motor memories with transcranial direct current stimulation allows later artificially-controlled retrieval" for further consideration at *eLife*. Your revised article has been favorably evaluated by Timothy Behrens as the Senior Editor, a Reviewing Editor, Rich Ivry, and one reviewer.

We appreciate your response to the initial reviews. We continue to think that this paper can make a novel and intriguing contribution to the literature on context effects in sensorimotor learning. However, we believe another round of revision is required. I think we can all agree that the effects are small here - not surprisingly so given the subtlety of the manipulation. Given this, we think it important that the primary analysis entail the fewest possible assumptions, and that these assumptions be thoroughly vetted in the paper.

The challenge you face in the analysis here is that your context effects are intermingled with natural decay processes. The curve fitting is designed to factor out the decay, but to do so requires making various assumptions (single vs double exponential), as well as entails lots of parameters since this is done on an individual basis for multiple epochs.

Your new Figure 3, which we think is a nice addition to the paper, suggests an alternative, and simple approach: use the S-T group data as a way to estimate the decay effect. These data are not involved in the critical analyses and thus constitute an independent method to look at decay, one that will not be conflated with the effect of stimulation context. In a sense, this provides a straightforward way to subtract out a baseline effect of the decay and then observe your main point of interest, the residual context effect. Making Figure 3 the key here, you end up subtracting 3d from 3b. In looking at things, it looks like the effect will be limited to the latter two probes. This would change your story to make the point that the effects of tDCS build up over time. We recommend that this analysis be the primary one in the manuscript, with panels A, C, and E providing the key raw data and the Panels B, D, and F the main point of comparison. You can still include the analysis from Figure 2, either moving this to a secondary analysis, or appended as an addition to a primary figure.

We were also confused why your model-free analysis was not performed in an analogous manner as you had done with the double exponential. You opted to turn to a bootstrap procedure here. While there is nothing wrong with the bootstrap approach, the lack of symmetry between the two approaches is confusing. It would seem more straightforward to employ a similar approach for the model-free and double exponential analysis and then presenting both analyses in the paper. Assuming the results converge on a common finding-namely that there is a residual context effect, the reader could then have confidence that the results were not specific to a given set of assumptions. As you now do things, we end up with a parametric analysis when considering a double exponential for decay and a non-parametric analysis when considering a single exponential.

---

## [Author Response]

*[…] While there is a general concern with the complexity of this analysis, there is an additional statistical issue here. Between subgroup differences in the fits may seem quite different for the anodal-cathodal switching pattern, but the patterns are not statistically independent. Fitting double exponentials may distort the estimate of the true anodal/cathodal switching effect. This is especially concerning if the switching effects are small compared to the size of the double exponential fits (as is the case here).*

*The reviewers believe that a much simpler analysis should be possible. The authors could simply analyze the differences in forces that are shown in the middle row, left panel of Figure 2. The middle panel (2e, difference for the sham condition) should probably be subtracted from the left panel (2b, the difference for the stimulation condition), and then the mean of each block can be statistically compared.*

We appreciate the reviewers for raising this important point. Our technique is guided by the fact that simple decay of motor memories should follow an exponential function (single or double) and that any deviation from such exponential should be due to the modulation of the force pattern by the polarity of the simulation. Therefore, we believe that calculating the residual by fitting the data with 2 exponential curves did not cause artificial stimulation-dependent modulation of the force but allowed us to eliminate the decaying component and slowly changing force output trends. Indeed, as shown in the figure below, the presence of force output modulation was not affected, even when fitting the data with a single exponential curve (see figure below and Figure 2—figure supplement 1 in the paper). The use of a double exponential curve function reduced the magnitude of the residuals, because it contributes to eliminate the slow drift in force with the data. Nevertheless, independently of the use of a single or double exponential function, we observed significant polarity-dependent modulation of the residuals, indicating strong evidence for the presence of this effect. The analysis of the residuals appears to us simpler than the bootstrap analysis suggested by the reviewers. In addition, it allows us to test for the existence of polarity-dependent changes in force output for each subgroup separately. Therefore, we took extra care in explaining this analysis into details and in describing the expected polarity-dependent changes in the residuals (third paragraph of the Results section, and Figure 2—figure supplement 1).

The reviewers are right in pointing that we need to make sure that the results that we obtained is not dependent on the model we use. According to the reviewers’ suggestion, we performed a model-free analysis of the force patterns during the test period. It should be noted that, as ∆Force was calculated from data recorded on different experimental groups, a bootstrap method was necessary to use the ∆Force for statistical testing. Therefore, rather than simplifying the analysis, the model-free approach makes it more complicated. Specifically, we created bootstrap samples of ∆Force (subtracting the CACA data from ACAC data for the T-T, S-T, and T-S groups) to obtain a mean and standard deviation of the resampled data (Figure 3 in the revised manuscript). We also created bootstrap samples for block-wise changes between 1^st^ and 2^nd^ blocks, 2^nd^ and 3^rd^ blocks, and 3^rd^ and 4^th^ blocks for the T-T, S-T, and T-S groups. Then, we tested the null hypothesis that there were no block-wise changes for the T-T, S-T or T-S groups (Figure 3). Additionally, we applied the same bootstrap method to test if block-wise modulation was greater for the T-T than for the S-T or T-S groups.

The results of bootstrap test was almost identical to the results on residual using ANOVA: The significant changes in ∆Force was observed from 1^st^ to 2^nd^ (p < 0.05), from 2^nd^ to 3^rd^ (p < 0.0001) and from 3^rd^ to 4^th^ block (p < 0.0001) for the T-T group. However, these changes were not observed for S-T and T-S groups except for the changes from 1^st^ to 2^nd^ block for S-T group. Furthermore, the changes were significantly greater for the T-T group than those for S-T and T-S groups (except from 1^st^ to 2^nd^ block between T-T and S-T groups). We believe that this additional analysis helps strengthen the results obtained with the residuals.

In the revised text, we have added results from this new analysis, as well the analytical procedure to the Methods section.

Author response image 1.Curves fitted for individual data (cyan and magenta) and the averaged curves (blue and red) when two exponential curves (**a**) and a single exponential curve (**b**) were used.The residual values were different (**c**, **d**), but the presence of modulation was not influenced.**DOI:**
http://dx.doi.org/10.7554/eLife.15378.014

[Editors' note: further revisions were requested prior to acceptance, as described below.]

*Thank you for resubmitting your work entitled "Tagging motor memories with transcranial direct current stimulation allows later artificially-controlled retrieval" for further consideration at eLife. Your revised article has been favorably evaluated by Timothy Behrens (Senior Editor), a Reviewing Editor, Rich Ivry, and one reviewer.*

*We appreciate your response to the initial reviews. We continue to think that this paper can make a novel and intriguing contribution to the literature on context effects in sensorimotor learning. However, we believe another round of revision is required. I think we can all agree that the effects are small here- not surprisingly so given the subtlety of the manipulation. Given this, we think it important that the primary analysis entail the fewest possible assumptions, and that these assumptions be thoroughly vetted in the paper.*

*The challenge you face in the analysis here is that your context effects are intermingled with natural decay processes. The curve fitting is designed to factor out the decay, but to do so requires making various assumptions (single vs double exponential), as well as entails lots of parameters since this is done on an individual basis for multiple epochs.*

*Your new Figure 3, which we think is a nice addition to the paper, suggests an alternative, and simple approach: use the S-T group data as a way to estimate the decay effect. These data are not involved in the critical analyses and thus constitute an independent method to look at decay, one that will not be conflated with the effect of stimulation context. In a sense, this provides a straightforward way to subtract out a baseline effect of the decay and then observe your main point of interest, the residual context effect. Making Figure 3 the key here, you end up subtracting 3D from 3B. In looking at things, it looks like the effect will be limited to the latter two probes. This would change your story to make the point that the effects of tDCS build up over time. We recommend that this analysis be the primary one in the manuscript, with panels A, C, and E providing the key raw data and the Panels B, D, and F the main point of comparison. You can still include the analysis from Figure 2, either moving this to a secondary analysis, or appended as an addition to a primary figure.*

*We were also confused why your model-free analysis was not performed in an analogous manner as you had done with the double exponential. You opted to turn to a bootstrap procedure here. While there is nothing wrong with the bootstrap approach, the lack of symmetry between the two approaches is confusing. It would seem more straightforward to employ a similar approach for the model-free and double exponential analysis and then presenting both analyses in the paper. Assuming the results converge on a common finding-namely that there is a residual context effect, the reader could then have confidence that the results were not specific to a given set of assumptions. As you now do things, we end up with a parametric analysis when considering a double exponential for decay and a non-parametric analysis when considering a single exponential.*

We greatly appreciate the detailed critiques of our paper. According to the reviewers’ suggestions, we now present the model-free analysis as the primary results in this new version of the paper.

We also seriously considered the reviewers’ suggestion to subtract the S-T data from the T-T data and perform an ANOVA on the residual data, as was done for the analysis of residuals. If the same participants had experienced both the T-T and S-T conditions, an ANOVA would have been appropriate. However, since T-T and S-T were different experimental groups in the present study, subtracted data could not be created for each participant. Thus, in order to perform the suggested ANOVA, the only possibility would be to subtract the averaged S-T data from each participant’s data for the T-T group.

However, we concluded that this approach would be problematic. First, an ANOVA on the residuals obtained as subtraction of averaged data of S-T group from each data of T-T group would not consider any variability in the S-T group. Second, since decay patterns differed from participant to participant, it would be difficult to determine what the residuals obtained by this way actually represent. Third, using the S-T group decay pattern as a baseline for the T-T group was not appropriate because each group received a different training protocol, which could influence the decay pattern (e.g., time constant). We believe the subtraction of the ACAC subgroup data from the CACA subgroup data (i.e., ∆Force) was more appropriate to eliminate the effect of natural decay because they received the same training protocol. It should be noted that since the data were collected from different groups, a bootstrapping (or permutation) method was necessary for the statistical test on ∆Force.

Nevertheless, we completely agree with the reviewers’ suggestion that contrasts between the T-T and S-T groups are quite important. These contrasts were already mentioned in the original manuscript; however, with this revision we tried to explicitly demonstrate them. Specifically, we have created a plot to demonstrate the difference in ∆Force between the T-T and S-T groups (Figure 4). We also created a plot for the T-T and T-S groups (Figure 4), and for the T-T and PPC groups (Figure 5). These new plots clearly indicate the presence of different modulation patterns between the T-T and other control groups. In this revision, we used a permutation test to statistically assess the effect of block order and period (please see Figure 2—figure supplement 1 for the definitions of block order and period; we have also revised the Statistics section to clearly explain the adopted method) on ∆Force and the changes from the 1^st^ to 2^nd^, from the 2^nd^ to 3^rd^, and from the 3^rd^ to 4^th^ block. Using a permutation test did not substantially change the results.

As for the absence of significant changes from the 1^st^ to 2^nd^ block for the T-T and S-T groups, we have now added a brief description (Results section, subsection “Active stimulation during the training and test periods is required for tagging and retrieval of motor memories”).